# Combining the Monthly Drought Code and Paleoecological Data to Assess Holocene Climate Impact on Mediterranean Fire Regime

**Marion Lestienne [1,2,*], Christelle Hély [2,3], Thomas Curt [4], Isabelle Jouffroy-Bapicot [1] and Boris Vannière [1,5]**

[1] Chrono-Environnement, CNRS, Université Bourgogne Franche-Comté, 25000 Besançon, France; Isabelle.Jouffroy@univ-fcomte.fr (I.J.-B.); boris.vanniere@univ-fcomte.fr (B.V.)

[2] The Institute of Evolution Sciences of Montpellier, Université Montpellier, CNRS, EPHE, IRD, 34090 Montpellier, France; christelle.hely-alleaume@umontpellier.fr

[3] Ecole Pratique des Hautes Etudes, PSL University, 75006 Paris, France

[4] National Research Institute of Science and Technology for Environment and Agriculture, Mediterranean Ecosystems and Risks, 13100 Aix-en-Provence, France; thomas.curt@irstea.fr

[5] MSHE Ledoux, CNRS, Université Bourgogne Franche-Comté, 25000 Besançon, France

* Correspondence: marionlest.jc@gmail.com

**Abstract:** Currently, indexes from the Fire Weather Index System (FWI) are used to predict the daily fire hazard, but there is no reliable index available in the Mediterranean region to be compared with paleofire records and check for their long-term reliability. In order to assess the past fire hazard and the fire-season length, based on data availability and requirements for fire index computation, we first chose and tested the efficiency of the Drought Code (DC) in Corsica (the main French Mediterranean fire-prone region) over the current period (1979–2016). We then used DC as a benchmark to assess the efficiency of the Monthly Drought Code (MDC) and used it to assess the Fire-Season Length (FSL), which were both used to characterize the fire hazard. Finally, we computed the Holocene MDC and FSL based on the HadCM3B-M1 climate model (three dimensional, fully dynamic, coupled atmosphere-ocean global climate model without flux adjustment) datasets and compared both index trends with those from proxies of paleofire, vegetation, and land use retrieved from sedimentary records in three Corsican lakes (Bastani, Nino, and Creno). Our strategy was to (i) assess fire hazard without the constraint of the daily weather-data requirement, (ii) reconstruct Holocene fire hazard from a climate perspective, and (iii) discuss the role of climate and human fire drivers based on the MDC-Paleofire proxy comparisons. Using both the Prométhée fire database and the ERA-Interim climate database over Corsica for the current period, we showed that DC values higher than 405 units efficiently discriminated fire-days from no-fire-days. The equivalent threshold value from MDC was set at 300 units. MDC and FSL indexes calculated for each of the past 11 millennia Before Present (11 ka BP) showed high values before 7 ka BP (above 300 units for MDC) and then lower values for the mid- to late Holocene (below 300 units for MDC). Climate appeared as a key driver to predict fire occurrences, promoting fires between 11 and 8 ka BP when summers were warmer than the current ones and reducing fire hazard after 7–6 ka BP due to wetter conditions. Since 5 ka BP, humans have taken control of the fire regime through agro-pastoralism, favoring large and/or frequent events despite less fire-prone climate conditions. The current fire hazard and fire-season length computed over the last few decades (1979–2016) both reported values that were respectively higher and longer than those assessed for the previous six millennia at least and comparable for those before 7 ka BP. For the next decades, due to climate warming associated with land abandonment (fuel accumulation) and the increase in human-related sources of ignition, we can expect an increase in fire hazard and larger fire events.

**Keywords:** fire hazard; fire weather index; drought code; fire-season; modelling; paleofire; fire driver; Corsica

## 1. Introduction

Fire is an integral part of ecosystems all around the world [1–4], and the average conditions in terms of fire seasonality, frequency, area burned, severity, and intensity define the given fire regime for each type of ecosystem [5]. Located at the interface between temperate European and subtropical North African conditions, the Mediterranean climate is characterized by a seasonal warm and dry climate with a marked summer drought [6,7], making it a fire-prone region [8,9]. For instance, during the last few decades (1979–2016), 9646 fires larger than 5 ha occurred in the French part of the Mediterranean Basin, and 830,566 ha (~22,447 ha·year$^{-1}$) have been burned [10]. Several factors influence the fire occurrence and behavior. The relative roles of climate, vegetation, and humans are still debated [11–13]. However, climate is probably one of the superordinate drivers of fires at regional scales [14] by controlling fire weather [15] (corresponding to the important factors determining fire probability of occurrence and fire behavior [16]), lightning-induced ignition [17], and the amount and distribution of flammable biomass [18,19]. Currently, humans are a superimposed driver by their footprints on the vegetation composition (with crops, pasture, and deforestation) [20] and their organization in the landscapes, or directly by igniting fires (accidents, negligence, or intentional ignitions) [21,22]. By considering the climate as one of the main fire drivers, it is likely that changes in past climate had a substantial effect on wildfire history [1,21]. Several studies used linear or non-linear models to relate meteorological variables to those of fires, and a large number of them used the indexes of the Fire Weather Index system (FWI) [23], which employs daily meteorological conditions to compute a variety of indexes aiming to estimate fuel moisture content, potential fire speed, and intensity, and an overall fire hazard index. While the FWI system was first developed for Canadian forests, it found worldwide applications, especially in the Mediterranean basin [24–26], where it has good results for predicting fire hazard [27]. However, no reliable index has ever been tested to characterize both past and present variables of fire regimes, mainly due to the difficulties to obtain daily past weather-data beyond the historical instrumental period. This study aims to test whether the Monthly Drought Code (MDC) [28], also originally developed for the boreal forest, could be used for the Mediterranean region. This is in order (i) to assess fire hazard accurately without the constraint of the daily climate-data requirement as only monthly means and the maximum are required, (ii) to reconstruct the Holocene (last 1700 years) fire hazard from a climate perspective using climate model simulations, and (iii) to discuss the respective roles of climate and humans as fire drivers.

The interest in testing the use of monthly data instead of daily data comes from the availability of climate model simulations and databases for past and future projections [29–31] and for historical observations [32], as they are mostly provided as monthly means. FWIs have been shown interest for the Mediterranean area to predict fire hazard associated with weather conditions [24–26]. Moreover, DC and MDC are very well correlated in the North American region [1] and are used to predict or to reconstruct fire hazard and fire events. From these two points, we hypothesized that MDC and the Fire-Season Length (FSL) could also be pertinent indicators for fire-weather condition prediction in the Mediterranean region.

In order to validate the changes in Holocene fire-weather conditions and based on the availability of past climate data, we computed the MDC values centered on each millennium for the past 11 millennia and compared them with regional paleofire reconstructions based on sedimentary charcoal contents from three Corsican lacustrine records (one as an original charcoal dataset presented here and two from the literature). The French island of Corsica is located in the western Mediterranean basin and is among the best preserved islands in the Mediterranean region in terms of plant diversity [33,34].

As in other areas of the Mediterranean region, the relative role of humans and climate in Corsican fire regime is still debated [9,35–37].

This study contributes to understand the underlying factors influencing fire regimes during the last millennia in the Mediterranean region both by providing a new high-resolution and well-dated charcoal record, which completes a previous study on Holocene plant diversity in Corsica [33,38], and by applying and testing an alternative fire hazard index adapted to past climate simulations and current observations.

## 2. Materials and Methods

### 2.1. Study Area and Sampled Lakes

Corsica is a French island located in the western Mediterranean basin, 80 km from the Italian coast and 160 km from the French coast (Figure 1). In spite of high mountains, there are no glaciers and no permanent snow on the island [39]. The vegetation was mainly composed of pinewood (*Pinus* sp.) and Ericaceous species (*Erica* sp.) during the early Holocene, before a significant change that occurred during the Neolithic (around 6 ka cal. Before Present (BP)), notably with the increase in oak forests and woodlands (*Quercus* sp.), which dominated the island during the remainder of the Holocene [33,40,41]. Currently, the high population and settlement density on the island and pastoral activity abandonment are very important stakes that, combined with the warming climate, induce a high fire risk [8,9].

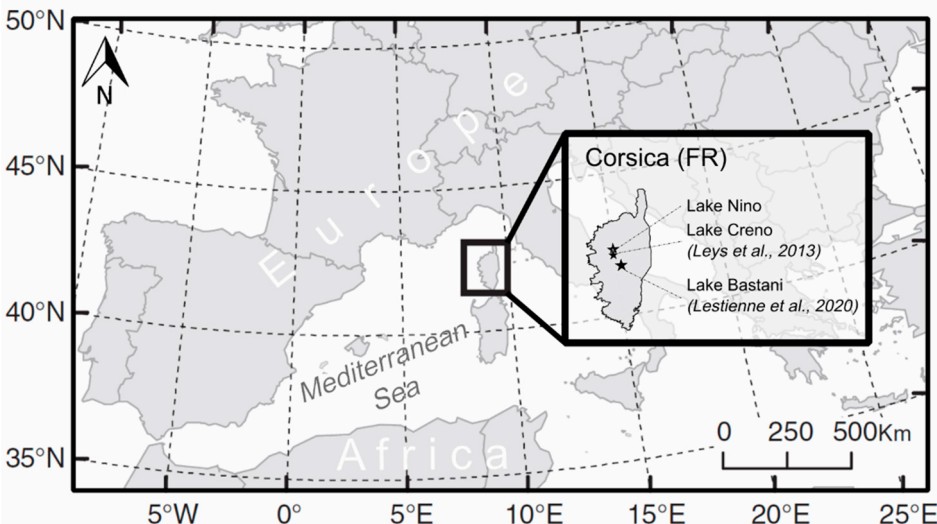

**Figure 1.** Geographical context of Corsica (France) and lakes from which sedimentary charcoals were extracted: Nino (the present study), Creno [42], and Bastani [33].

Sediment cores from three mountain lakes were studied for fire reconstructions in Corsica, namely Lake Bastani [33], Lake Creno [9], and Lake Nino (this study) (Figure 1) [43]. Lake Bastani (42°06′ N, 9°13′ E) is one of the most elevated lakes in Corsica. Because its watershed is small and due to the topography, this windward lake is a good captor for wind-transported particles including charcoals produced in the surrounding regional area [44,45]. Conversely to Lake Bastani, Lake Nino (42°25′ N, 8°94′ E, 650,000 m$^2$) and Lake Creno (42°12′18′′ N, 08°56′45′′ E) (Figure 1) are lower in altitude and are expected to capture more local fires. A detailed study for Lake Bastani, including vegetation and fire histories over the last 11 ka BP based on pollen assemblages and charcoal influx, was presented in Lestienne et al. [33]. The complete study for Lake Creno, including fire change reconstructions over the last 10 ka BP based on charcoal influx, was presented in Leys et al. [42]. The present study combined the fire reconstruction from Bastani [33] and Creno [42] with the original paleofire record from Nino, and all are presented here for comparison with the monthly drought code index computed throughout the Holocene.

## 2.2. Charcoal, Pollen, and Fungal Remains Analyses

High-resolution and well-dated sedimentary records (see Figure A1 for the datation procedure and the resulting age-depth model) of the lakes Bastani, Nino, and Creno were used to reconstruct the past fire signal and the surrounding environment.

A total of 271 (for Bastani) and 618 (for Nino) contiguous sediment samples were retrieved along cores every 10 or 5 mm (depending of the sedimentation rate). For charcoal extraction, each sample was washed on an 80 μm mesh sieve after hydrochloric acid and hydrogen peroxide treatments according to the standardized macro-charcoal sieving method [46,47]. All charcoal particles from each sample were observed with a digital microscope coupled to the high-speed camera, Keyence VHX-5000 (Keyence Corporation, Ōsaka, Japan). Images, observed using a $100 \times$ magnification, were assembled to observe the entire sample on one picture with a high precision. From this picture and adding information on charcoal visual characteristics (color and brightness ranges), the microscope software performed a semi-automated counting of charcoal particles present in the sample: each particle corresponding to the color and brightness ranges chosen was selected, and the user checked each particle visually. For Creno, charcoal particles were counted by the authors from 891 continuous samples using similar digital image analysis software (WinSeedle 2007, © Regent Instruments Inc., Quebec City, Canada) [9], and the published reconstruction with its own age-depth model can be compared to Lake Nino and Lake Bastani paleofire records.

A total of 21,138 (Bastani), 38,622 (Nino), and 2705 (Creno) charcoal particles were identified. The charcoal record was quantified by calculating the CHarcoal Accumulation Rates (CHAR), i.e., the quantity of charcoal particles per volume of sediment and per unit of time according to the sedimentation accumulation rate estimated by the depth–age model ($\#/cm^2/year$). Based on the sample-age average, the cores were resampled by 20 years (approximately the mean step of Nino and Creno cores) to make them comparable.

A smooth curve was generated from the resampled and rescaled (Z-score) CHAR values of the three lakes (Bastani, Nino, and Creno) using the LOESS (Locally-Estimated Scatterplot Smoothing) method.

Dealing with other proxies available from Bastani [33], the AP/NAP ratio contrasted the total number of Arboreal Pollen grains (AP) to the Non-Arboreal Pollen grains (NAP) and was a surrogate for the vegetation land cover (i.e., woody versus non-woody cover) [48]. Crop and ruderal pollen taxa were part of NAP and were indicators of human presence [49]. Fungal remains, and particularly *Sporormiella* sp., were local dung indicators and, in turn, a pasture marker, and therefore a human activity marker [50]. The detailed method and results for charcoals, pollen, and fungal remains analysis are available in Lestienne et al. [33].

## 2.3. Current Climate and Fire Datasets

Current climate data were extracted from ERA-Interim [32], which is a global atmospheric reanalysis from 1979, continuously updated in real time, with a spatial resolution of approximately 80 km, Corsica being covered by a total of five pixels [32] (Figure A2).

Current fires' data were extracted from the Prométhée database [51], which includes the date of ignition, origin (anthropogenic or naturally ignited), size, and location ($2 \times 2$ km resolution) of wildfires that have occurred since 1973 in southern France. To assess the fire regime, we only extracted Corsican wildland fires larger than 5 ha in size and from 1979 to 2016 to match with the ERA-Interim time period (Table 1).

**Table 1.** Main characteristics of the current Corsican fire regime and climate normal over the 1979–2016 period. Fires (>5 ha) were extracted from the Prométhée database [51], and climate data were extracted from ERA-Interim Climate Models [32].

| Dataset | Variable | | Unit | Value |
|---|---|---|---|---|
| Fire regime | Fire frequency | | #·year$^{-1}$ | 84 |
| | Burned area | max | ha | 5644 |
| | | median | | 10 |
| | | mean | | 82 |
| | Fire Season | | month | June to September/October |
| Climate data | Mean temperature (temp) | autumn | °C | 16.2 |
| | | winter | | 12.1 |
| | | spring | | 20.1 |
| | | summer | | 26.4 |
| | Mean precipitation (prec) | autumn | mm·month$^{-1}$ | 99 |
| | | winter | | 57 |
| | | spring | | 45 |
| | | summer | | 27 |
| | Relative humidity (rh) | autumn | % | 69.5 |
| | | winter | | 66.2 |
| | | spring | | 54.7 |
| | | summer | | 50.9 |
| | Wind speed (ws) | autumn | km·h$^{-1}$ | 12.8 |
| | | winter | | 13.6 |
| | | spring | | 11.5 |
| | | summer | | 9.7 |

### 2.4. Climatic Model

Paleoclimate conditions were extracted from HadCM3B-M1 simulations (three dimensional, fully dynamic, coupled atmosphere-ocean global climate model without flux adjustment) [51], this model being a variant of the fully complex Hadley Centre Climate Model HadCM3, usually involved in the Intergovernmental Panel on Climate Change assessment reports. The HadCM3B-M1 variant was originally the most commonly used [52,53]. It is a three-dimensional, coupled atmosphere–ocean global climate model without flux adjustment, which is very similar to that described by Gordon et al. [54]. This model performed a snapshot equilibrium simulation per millennium, from which climate normals (monthly means) for main variables (i.e., air temperature and precipitation used here) were computed (Figure A3). We therefore used these datasets of climate normals for the last eleven millennia (centered at 11, 10, ... , 2, and 1 ka BP), as well as for the control run representative of the pre-industrial period (i.e., AD ca. 1750, considered as equivalent to 0 BP) to compute temperature and precipitation anomalies (i.e., difference and rate of change for temperature and precipitation, respectively) for each millennium [55].

Using an inverse-distance weighting approach on the four closest pixels from those covering Corsica, we downscaled these anomalies by applying them to the current period climate normals computed from the ERA-Interim dataset (80 × 60 km resolution, 5 pixels used to include all of Corsica) to reconstruct past climate for each Holocene millennium at the whole island scale [55,56].

### 2.5. Fire Weather Index System

The Fire Weather Index System is part of the Canadian Forest Fire Danger Rating System, which has been developed by Forestry Canada since 1968 [23,57]. It is a weather-based system that models fuel moisture using a dynamic bookkeeping system that tracks the drying and wetting of distinct fuel layers in the forest floor, i.e., their potential flammability (Figure A4). The first three moisture

codes of the FWI represent the moisture contents of three superposed humus layers: the surficial layer fine fuels "Fine Fuel Moisture Code" (FFMC, 1–2 cm deep), the loosely compacted duff layer with organic material "Duff Moisture Code" (DMC, 5–10 cm deep), and the deep duff layer of compacted organic material "Drought Code" (DC, 10–20 cm deep). The DMC and DC indexes were combined to create a generalized index of the availability of fuel for consumption "BUildup Index" (BUI), while the FFMC was combined with the wind speed to estimate the potential fire rate of spread "Initial Spread Index" (ISI). Finally, the BUI and ISI were combined to create the FWI, which represented the potential intensity of a spreading fire, and therefore the overall fire hazard [23].

The DC was calculated with the R software (cffdrs package) and was a function of the "Daily Humidity index" (DH) and the Potential Evapotranspiration (PE) [23]; see Figure A4 for the computation details.

Daily climatic data of the whole island from ERA-Interim were used to calculate the FWIs during the current period (1979–2016). However, we aimed to calculate an index usable for the entire Holocene, and wind speed and relative humidity changes over long periods may be less reliable when extracted from climate model simulations due to their intrinsic daily and subdaily variabilities. Therefore, we chose to use precipitation and temperature only, and consequently to focus exclusively on the drought code [23]. Its time lag for complete drying was 52 days, so it indicated the effects of seasonal drought on forest fuels and the probability of smoldering in deep duff layers and in large logs. It was a simple moisture bookkeeping system that used an estimate of daily temperature to estimate a day's potential evapotranspiration, following the method of Thornthwaite and Mather [58], and daily rainfall to track increases in wetness of the deep layer (Figure A4). The fact that there was no human activity related to the DC computation was a valuable characteristic that allowed us to use it to assess past periods such as earlier in the Holocene or before. DC values between days with and without fires were compared over the 1979–2016 period to find a DC threshold value that would characterize fire-days (i.e., a value assumed excluding at least 75% of the no fire-days and including at least 75% of the fire-days). Then, we tested the efficiency of the monthly drought code [28] by comparing it to DCmean. DCmean simply represents the monthly mean of the daily values of DC, while MDC was originally created to be computed from monthly means of precipitation and maximum temperatures [28]. For Canada, an MDC value higher than 280 units has been associated with an extreme drought and corresponds to a high fire hazard [28]. We therefore searched for a threshold value allowing detecting months with high fire hazard based on Corsica MDC values computed with the ERA-Interim database over the 1979–2016 period. Then, MDC values were computed for the entire Holocene (0 to 11 ka BP) based on the HadCM3B-M1 temperature and precipitation datasets, and their trend compared with the paleofire trend from charcoal records. The MDC is not intended to be used in operational situations where daily weather data are available for fire managers, but it could be very helpful to highlight past or future changes in droughts and consequences on fire hazard [28,59].

*2.6. Fire Season Length*

Based on historical analysis of MDC values, and according to the method originally developed by Hély et al. [56] for DC, but adapted here to MDC, we used the MDC threshold value (presented in the Results Section) to calculate the fire-season length for the 1979–2016 period and for each Holocene millennium as well. Basically, for all months with MDC values above the threshold, we considered the full month length (30 or 31 days) as part of the fire-season. To define the starting and ending months of the fire-season and to add the related number of days (i.e., < 30 or 31 for each month), we used a basic linear interpolation between each of these months reporting MDC values lower than the threshold value, but just preceding or following a month with above-threshold MDC values.

## 3. Results

### 3.1. DC and MDC Efficiently Detect Fire Days/Months of the Current Period (1979–2016)

As compared to the other indexes of the fire weather index related to fuel moisture (Figure A5), DC performed better as it significantly discriminated days with and without fires (Figure 2). Fire days from the Prométhée database showed median and mean DC values that were ca. 300 units higher (572 and 548, respectively) than DC values recorded during days for which there were no fires reported (166 and 250, respectively). Moreover, the larger the fire, the higher DC (Figure 2). In order to exclude most of the no-fire-days without excluding too many fire-days, we conservatively defined the DC value of 405 units (75% no-fire-days were excluded, while 78% of fire-days were conserved) as the threshold above which days were considered as potential fire-days and also considered as belonging to the fire-season.

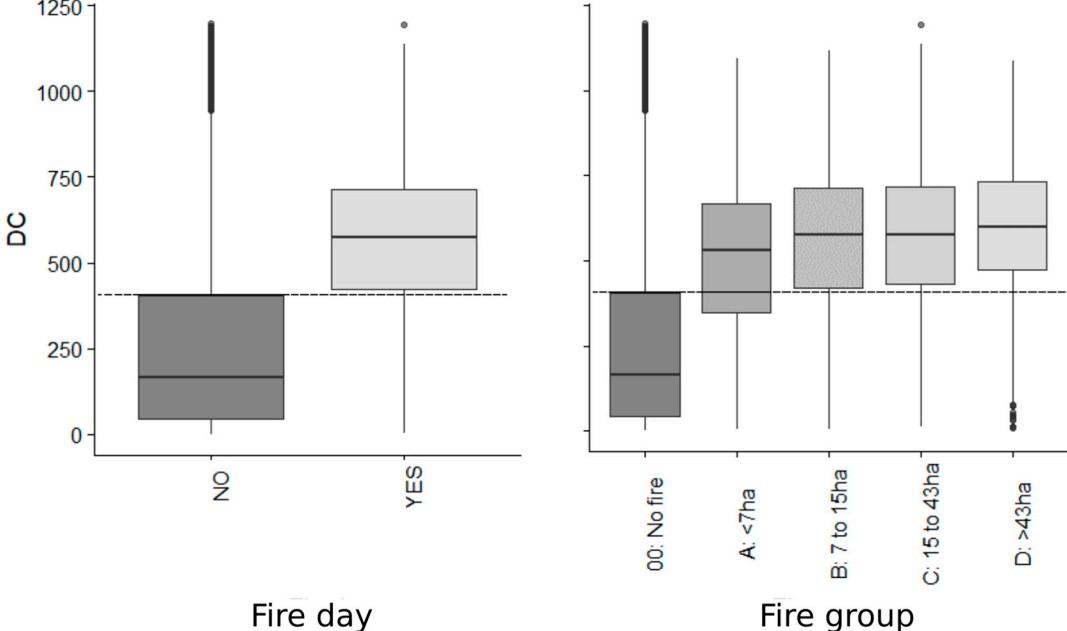

**Figure 2.** Comparisons of Drought Code (DC) distributions between days with or without fire reported (left panel, *t*-test, *p* > 0.001) and among increasing fire size classes (defined from the distribution quartiles) as compared to the no-fire class (right panel, first class). For both panels, fires and related day conditions were extracted from the Prométhée database. The horizontal dotted line in each panel corresponds to the chosen DC threshold (405 units) in order to calculate the fire-season length. It excludes 75% of the no-fire-days and includes 78% of the fire-days.

Monthly means of daily DC (i.e., DCmean) over the 1979–2016 period and for each month of the fire-season (April–October) were compared to the MDC values computed over the same months and period. DCmean values were slightly higher than MDC values, yet their relationship (linear regression model) and their covariation (non-parametric Kendall rank correlation coefficient) were highly significant (Figure 3a, MDC = 0.72 × DC). Moreover, the intra-seasonal monthly trends reported by both metrics were very similar (Figure 3b), suggesting the use of MDC as a good proxy of DC and therefore of the fire hazard to be tracked. Using the linear relationship between DCmean and MDC (Figure 3a), we inferred the 300 unit MDC as the threshold to be used to compute FSL and to analyze its change through the Holocene.

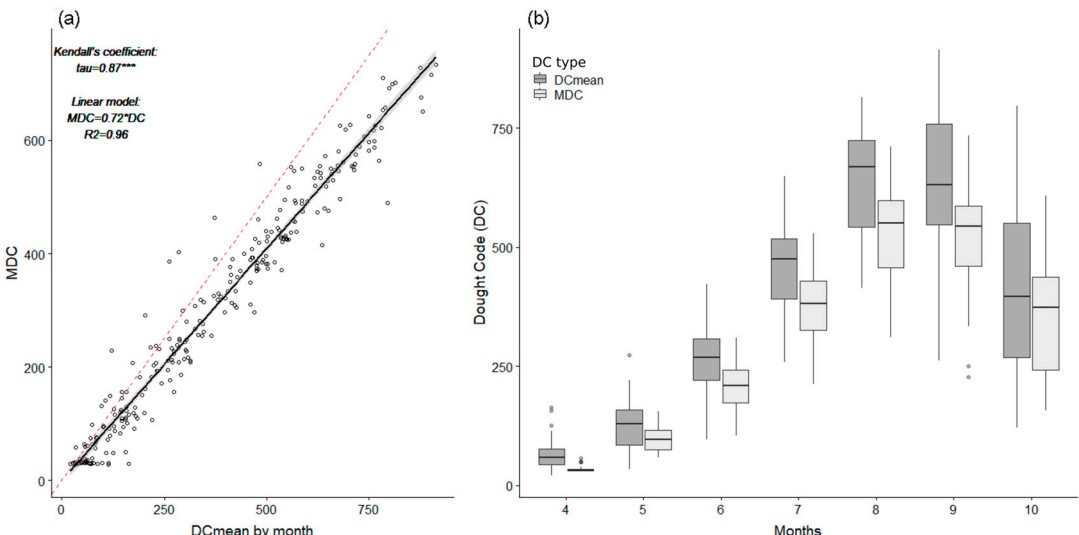

**Figure 3.** (**a**) Correlation between the Monthly Drought Code (MDC) and DCmean using monthly values from April to October over the 1979–2016 period (n = 532 corresponding to seven months each year for 38 years). The dotted red curve represents the perfect match between DCmean and MDC, while the black curve is the linear regression model found (MDC = aDC+b), and for which the normal distribution and homoscedasticity of residuals were tested.;(**b**) DCmean and the MDC changes along the fire-season (currently encompassing months from April to October).

The relationships over the 1979–2016 period between MDC values for each fire-season month and the monthly burned area or the monthly number of fires (Figure A6) were analyzed to test the efficiency of the MDC threshold efficiency in discriminating fire-prone months. Both burned area and number of fires were significantly correlated with the MDC values. Moreover, the 300 unit threshold discriminated the most fire-prone months (i.e., with more than 200 fire occurrences and/or more than 5000 ha burned).

### 3.2. Change in MDC and FSL during the Holocene

The summer MDC values were particularly high at the beginning of the Holocene (11 ka BP −9 ka BP) (Table 2, Figure 4), with the highest value of 685 units at 9 ka BP. Then, MDC decreased until 6 ka BP and remained stable (i.e., around 400 units) over the five next millennia, the MDC value at 1 ka BP being the lowest MDC value over the last eleven millennia (314 units). Finally, a new increase in MDC occurred, up to 419 units for the current period. Currently, the calculated fire-season starts in June and ends in October, lasting 111 days on average (Table 2). Over the entire Holocene, the fire-season length ranged from 54 to 126 days, with the longest fire-season from 11 to 7 ka BP (from 106 to 126 days, respectively). As now, during this early Holocene period, the fire-season started in June and ended in October. The sharp decrease in fire-season length between 7 and 6 ka BP represented a delayed start (in July), while the length stabilized around 70 days. Afterward, it shortened again between 2 ka BP and 1 ka BP, reaching the shortest duration (64 and 54 days, respectively). At that time, the fire-season both started later and ended earlier (i.e., from July to September, respectively).

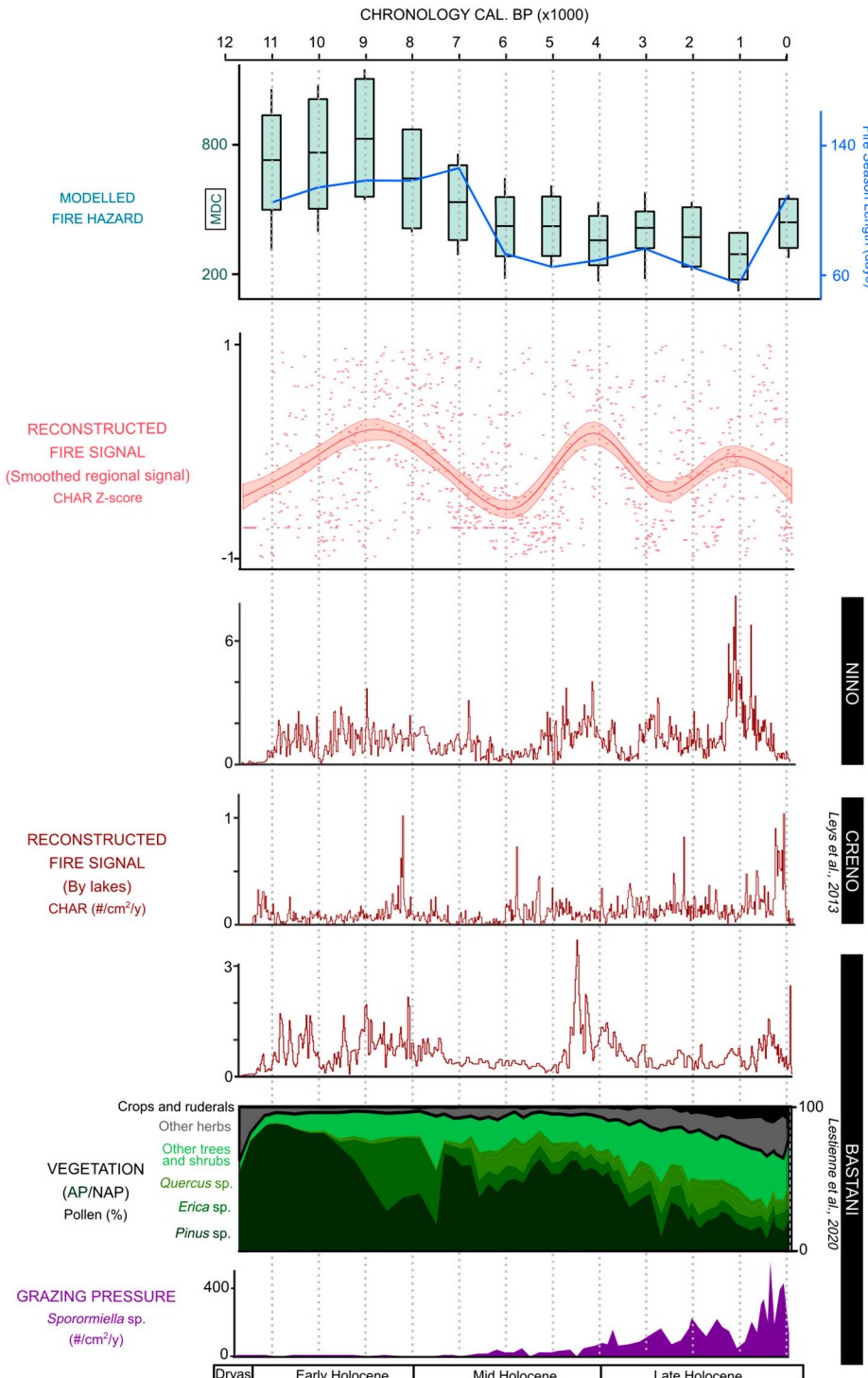

**Figure 4.** Changes in Monthly Drought Code (MDC), Fire Season Length, fire signal (from CHarcoal Accumulation Rates (CHAR)), vegetation (from pollen composition), and pastoral activities (from fungal remains) during the Holocene. MDC and fire-season length were computed at the island scale, while charcoals, pollen, and fungal remains were extracted from lake sediments. Creno data were extracted from Leys et al., 2013 [56], Bastani data from Lestienne et al. (2020) [33], and Nino data are original. BP, Before Present; AP, Arboreal Pollen grains; NAP, Non-Arboreal Pollen grains.

**Table 2.** Holocene characteristics in terms of summer fire hazard and fire-season length computed using HadCMB3-M1 simulation datasets [52].

| ka BP | Mean MDC | Fire-Season | | |
|---|---|---|---|---|
| | | **Starting** | **Ending** | **Length** |
| 0 | 419 | 11 June | 2 October | 111 |
| 1 | 314 | 21 July | 15 September | 54 |
| 2 | 379 | 26 July | 30 September | 64 |
| 3 | 394 | 22 July | 9 October | 77 |
| 4 | 365 | 21 July | 1 October | 70 |
| 5 | 412 | 28 July | 4 October | 66 |
| 6 | 406 | 23 July | 5 October | 72 |
| 7 | 482 | 3 June | 9 October | 126 |
| 8 | 556 | 13 June | 10 October | 117 |
| 9 | 685 | 18 June | 16 October | 118 |
| 10 | 628 | 17 June | 10 October | 113 |
| 11 | 598 | 18 June | 4 October | 106 |

*3.3. Holocene Paleofire and Environmental Changes from Lacustrine Sediment Reconstructions*

Over the Holocene, lacustrine sediment cores of Creno [42], Bastani [33], and Nino showed similar trends of temporal fire signal recorded by charcoals (Figure 4). From the point of view of fire hazard (i.e., MDC and FSL) and fire signal (charcoal records), we highlight two phases. The first phase lasted from 11 ka BP to 5 ka BP. During this phase, we observed that changes in both fire signal and fire hazard values were synchronous: the highest fire signal was observed when both the fire hazard was high and the fire-season long, and vice versa. The very early Holocene (before 11 ka BP) recorded almost no fire as compared to later on. This was followed by a period characterized by a strong increase in the fire signal, which matched with the period of highest fire hazard values. Then, a decrease in both signals was observed. The second phase spanned over the last five millennia. It was characterized by a new increase in fire signal after 5 ka BP, such increasing trends not being observed in the fire hazard and fire-season values, which stayed very low and short, respectively. Even though a decrease was observed between 3 and 2 ka BP, the fire activity stayed high until currently.

The synthesis of the palynological and fungal remain analyses of Lake Bastani showed that before 11 ka BP, the AP/NAP was low, meaning that the herbaceous taxa were dominant (Figure 4). Then, the ratio increased and stayed stable until 7 ka BP with the dominance of *Pinus* sp., followed by *Erica* sp. from 9 ka BP. The establishment of *Quercus* sp. and the increase in *Sporormiella* sp. have occurred since 7 ka BP, while the first synchronous increase in both crops and ruderals taxa and *Sporormiella* sp. influx has occurred since 5 ka BP.

## 4. Discussion

*4.1. The Efficiency of DC and MDC to Target Potential Fire-Days/Months and Assess the Fire-Season Length*

Beyond the ability of the FWIs that have been previously demonstrated for several Mediterranean countries [21,25,26,60–62] to predict the fire hazard on time [63–68], our results suggested their valuable use, especially the DC index, for detecting current days with or without fires and assessing ranges of fire sizes larger than 5 ha. The DC, with both its 405 unit threshold and its simplest computation only relying on precipitation and temperature, appeared to be therefore the best fire index candidate to monitor the probability of fire occurrences in the Mediterranean region when daily conditions were available. Moreover, its threshold value allowed both estimating the fire hazard based on the daily DC values and computing the length of the active fire-season and, in turn, its onset and termination dates.

Our results also showed that when daily conditions were not available (for past or future periods for which only monthly reconstructions or projections were stored), the MDC index was a very good surrogate of DC to assess fire hazard for the Mediterranean region. Indeed, the highly significant and

strong correlation found between DC and MDC was even higher than that for the boreal region for which it had been created originally [28], and later on applied [1]. For MDC, a 300 unit threshold was determined and successfully tested to detect the most fire-prone months. Another way to validate the use of MDC was the similar length of the fire-season calculated with DC versus MDC (105 versus 111 days, respectively) and the fact that such inferred lengths fit well with the current reported fire-season length of ~ 100 days observed in the Mediterranean region [26,62].

*4.2. Holocene History: From Climatic to Anthropogenic Fires*

The early Holocene (11 ka BP) was characterized by the post-glacial recolonization of the woody vegetation [40,41], well represented by *Pinus* sp. in our records. The record of many charcoal peaks (Figure 4) suggested a sustained fire regime in terms of fire frequency and biomass burned. This trend was observed in the sediments of the three lakes (with a 500 year earlier start for Lake Creno [9]), suggesting a regional trend. In the same time, MDC was high, which fit well with the dry summer conditions reconstructed based on fossil chironomid by Samartin et al. for Tuscany (Italy) [69]. The large amplitude of the values, added to the temperature and precipitation anomalies (Figure A3), illustrated the accentuated seasonality due to insolation increase [6,69,70]. These results were in line with other studies, which described the warmer condition of the early Holocene [71–73] and confirmed our previous hypothesis [33], which stated that climate, adding to the increase in the fuel availability, may explain Corsican fires between 11 ka BP and 7 ka BP [40,74]. The fire signal recorded was quite similar between Lake Bastani and Lake Nino during this period, with similar peak values (e.g., 9 and 8 ka BP) and low values (e.g., 10 ka BP). The similarity existed, but was less obvious for Lake Creno, probably due to the overall lower charcoal signal recorded for this sequence [9]. This strengthened the idea of a regional trend dominated by many fire events occurring both in Corsica and elsewhere over the Mediterranean basin and adjacent regions during the early Holocene [12,75].

The period starting at 7 ka BP highlighted a simultaneous strong decrease in charcoal peaks (in particular for both Bastani and Creno) and a sharp change in vegetation composition, with the decrease in *Pinus* sp. and *Erica* sp. and the increase in *Quercus* sp. [40,74]. Such vegetation change was simultaneous to a strong decrease in the MDC values and the shortening of the fire-season length, likely due to wetter spring conditions (Figure A3). Such a wetter climate after 7 ka BP has been reported in several studies [9,40,76], but our results, combined with a high-resolution vegetation record, permitted highlighting the rapidity of this change and its targeted season, and so, to attribute this major event to a climatic cause. The stability of charcoal signal, MDC, and fire-season length until 5 ka BP showed that the ecosystem reached a new equilibrium more adapted to these wetter climate conditions.

A significant increase in *Sporormiella* sp. spores indicated the presence of large herbivores around the lake, at least since 5 ka BP. The frequentation of the lake area by livestock seemed to be a plausible explanation of such dung fungal spores increase. Moreover, these clues of human presence were followed by an increase in charcoal content for both Bastani and Nino (and a bit earlier at Creno), which could be interpreted as land use transformation into crops and pastures and the associated deforestation [77,78]. That was in good accordance with archaeological knowledge. The Chalcolithic period in Corsica with megaliths and fortified habitats was considered as a population growth period [79]. The increase in biomass burning reduced the tree proportions and opened the landscape, as attested by the AP/NAP ratio decrease, particularly marked since 3 ka BP. This forest opening during the late Holocene has also been observed in Sardinia [80], Iberia [6], and southern France, including Corsica at Creno [40]. Moreover, our results showed a clear increase in anthropogenic activities' indicators such as pollen of ruderal and crop species and a clear increase in pasture indicators such as *Sporormiella*. In agreement with other studies [12,80,81], we mainly attributed this opening (increase in NAP and cultural indicators) and the increase in fire events to human activities (crops and pasture), also attested on others Mediterranean islands like Sardinia [80], Sicily [82], or Majorca [83]. The stability of a relatively low MDC and a short fire-season were arguments to suggest that the main driver for fire activity had shifted from climate to humans since 5 ka BP.

A new decrease in MDC occurred between 2 ka BP and 1 ka BP, reaching the lowest values of both MDC and fire-season length over the Holocene, then followed by a strong increase in grazing indicators and charcoal peaks still indicating that human activities contributed to increase fire frequency [12]. These human activities, in particular crops and pasture, affected all ecosystems, from the subalpine pastoralism to the Mediterranean olive groves [33]. Humans have opened more and more the landscape up to the current Corsican landscape. This period of demographic increase is probably linked to the strong Tuscan immigration into Corsica around 1 ka BP [79] and has contributed to increased fire frequency over the last few centuries.

The strong similarities between the three lakes attests to their good quality as paleofire recorders at a regional scale. Their records have the same early Holocene history, which attests that a common driver, probably climate, controlled the entire Corsican fire regime. The differences observed after 6 ka BP should be the results of local events and slightly different human histories occurring everywhere in Europe from this period [12,75,77,84–86].

### 4.3. The Current Climate Is Getting Close to the Mid-Holocene Climate Conditions

The early Holocene climate is known to have been dry and warm, with high seasonality (i.e., strong differences between summer and winter) [71–73,87]. Our results confirmed a climatic shift staged between 7 ka BP and 5 ka BP. During this transitory period, MDC decreased below 400 units, and the fire-season became shorter (less than 100 days). However, the most recent value of MDC (corresponding to the 1979–2016 period) showed an increase, and this increase was associated with a strong increase in the fire-season length resulting from an earlier onset in June. For the first time since 5 ka BP, the fire-prone climatic conditions (i.e., MDC combined with the fire-season length) were closer to those from the early Holocene, and these conditions are favorable for fire ignitions and propagation [8,9]. Moreover, the last few decades were marked by a decline of pastoral activities and an increase in land abandonment, causing a closure of the environment and fuel accumulation [88,89]. This closure combined with the global warming, recorded in our results, could promote future uncontrolled fire episodes [26,90–92].

### 4.4. Limitations of the Study

The calculation of MDC needs only two variables (temperature and precipitation), and changes in vegetation or human pressure on ignitions are not considered to assess changes in the overall fire hazard. However, through its composition, density, and spatial arrangement, vegetation directly influences the type of fire and its characteristics, e.g., [80,93,94]. Currently, a high fire hazard is correlated to a fire-day (DC) or a fire-prone month (MDC). However, past vegetation was different and could have promoted more or fewer fires. A way to improve the robustness of this method should be to test it in different landscapes with different vegetation. Finally, if we saw that DC was able to discriminate fire-days and that MDC can discriminate fire-prone months, it is indeed important to understand that today's fires are mostly linked to human activity. From this logic, it is likely that the threshold value chosen on the basis of current data overestimated the fire danger over the oldest periods.

## 5. Conclusions

In this study, we tested first the efficiency of DC to discriminate fire-days from no-fire-days over the present (1979–2016 period) and the efficiency of MDC, which is a simplified version of DC, to discuss climate vs. other fire drivers during the Holocene. MDC was used here for the first time for both a Mediterranean region (Corsica) and the entire Holocene. Combined with three paleofire records obtained from sedimentary charcoals' quantification and the vegetation and human activities' dynamics reconstructed from pollen and fungal remains, MDC permitted pointing out the drivers of fire history in Corsica for different periods. Firstly, the dry and warm summer conditions induced frequent fires and important biomass burning before 7 ka BP. Then, the wetter conditions induced a decrease in fire frequency, which allowed long-term post-fire succession and, in turn, a closure of the

forest. Finally, from 5 ka BP, humans might have been the main driver of vegetation dynamics and of fire occurrences by deforesting and developing crops and pastures using fire, despite a lower fire hazard and a shorter fire-season due to wetter and therefore less fire-prone climate conditions.

MDC appeared as a simple, but efficient complementary tool to go back into the past and to understand the underlying factors of fires by reconstructing the Holocene climate-related fire hazard. It allowed avoiding the difficulties in acquiring and/or simulating daily data, and it improved our understanding of wildfire hazard metrics (including fire-season length) at a regional scale. Nevertheless, it is necessary to be aware of other drivers (i.e., humans and vegetation types) in order to understand the complexity of fire regime. The modern decrease in pastoral activities associated with land abandonment, combined with the increase in fire frequency and intensity expected in the next few decades due to global warming and human density increase, threaten most European landscapes. The current fire hazard and length of the fire-season are, for the first time for millennia, as high and long, respectively, as those of the early Holocene period. The next step would be to calculate MDC values from future climate scenarios in order to assess the future trend of fire hazard.

**Author Contributions:** Conceptualization, C.H., B.V. and M.L.; methodology, validation, investigation, resources, writing—review and editing and visualization, M.L., C.H., T.C., I.J.-B. and B.V.; formal analysis and data curation, M.L., C.H. and I.J.-B.; writing—original draft preparation, M.L.; supervision, B.V. and C.H.; project administration and funding acquisition, B.V.. All authors have read and agreed to the published version of the manuscript.

**Funding:** This research was funded by Région Bourgogne Franche-Comté through Chrono-environnement laboratory, the MSHE (Maison des Sciences de l'Homme et de l'Environnement) Ledoux and the projects ONOMAD (Optimisation Numérique des Observations Microscopiques), QMedFire (Quantification et modélisation des régimes des feux méditerranéens), and ENVILEG (Héritage environnemental des grandes transitions socio-écologiques en Europe) led by Boris Vannière. This study was also supported by the CNRS PaléoMEx-MISTRALS (Mediterranean Integrated STudies at Regional And Local Scales) program. This study is part of the PAGES-GPWG (PAst Global changES – Global Paleofire Working Group) activities.

**Acknowledgments:** We thank the Environmental Office of Corsica, the municipalities of Ghisoni, Corte, and Casamaccioli, and the Regional Natural Park of Corsica for the lake access permit. We thank also Bérangère Leys for giving us full access to Lake Creno charcoal data. We thank the 3 anonymous reviewers for contributing to the improvement of the manuscript.

**Conflicts of Interest:** The authors declare no conflict of interest.

## Appendix A

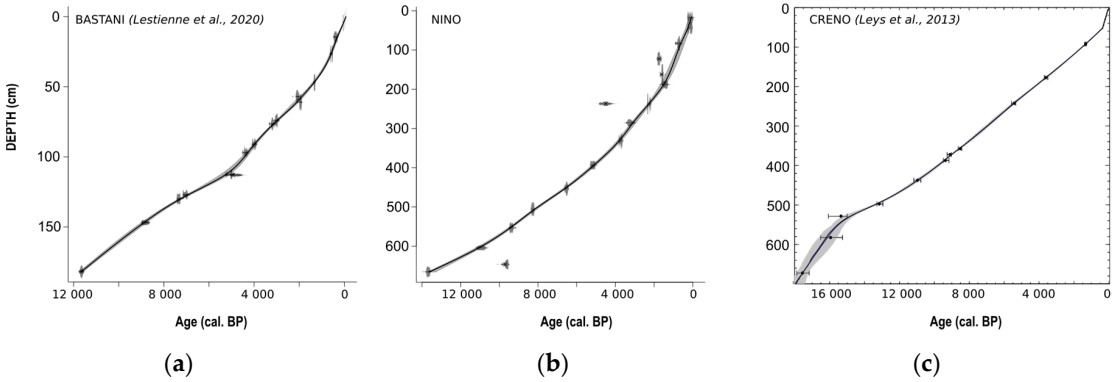

**(a)**                                                      **(b)**                                                      **(c)**

**Figure A1.** Age-depth model for Bastani [33], Nino, and Creno [42] sequences. The chronological control was based on a combination of (**a**) 10 (Bastani)/18 (Nino)/11 (Creno) radiocarbon dates obtained at the Poznan Radiocarbon Laboratory on diverse macro-remains of terrestrial origins (leaves, seeds, charcoals, and wood). (**b**) One age estimated for the main late-glacial/early-Holocene transitions indicated by the pollen stratigraphy and the geochemistry data (Bastani). (**c**) Two (Bastani)/seven (Creno) radiometric markers derived from short-lived radionuclides (210Pb, 137Cs) for the top of the core. Based on these chronological markers, the Clam package (R software, R. Core Team, 2018) was used to generate an age depth model within the 95% confidence limits [95].

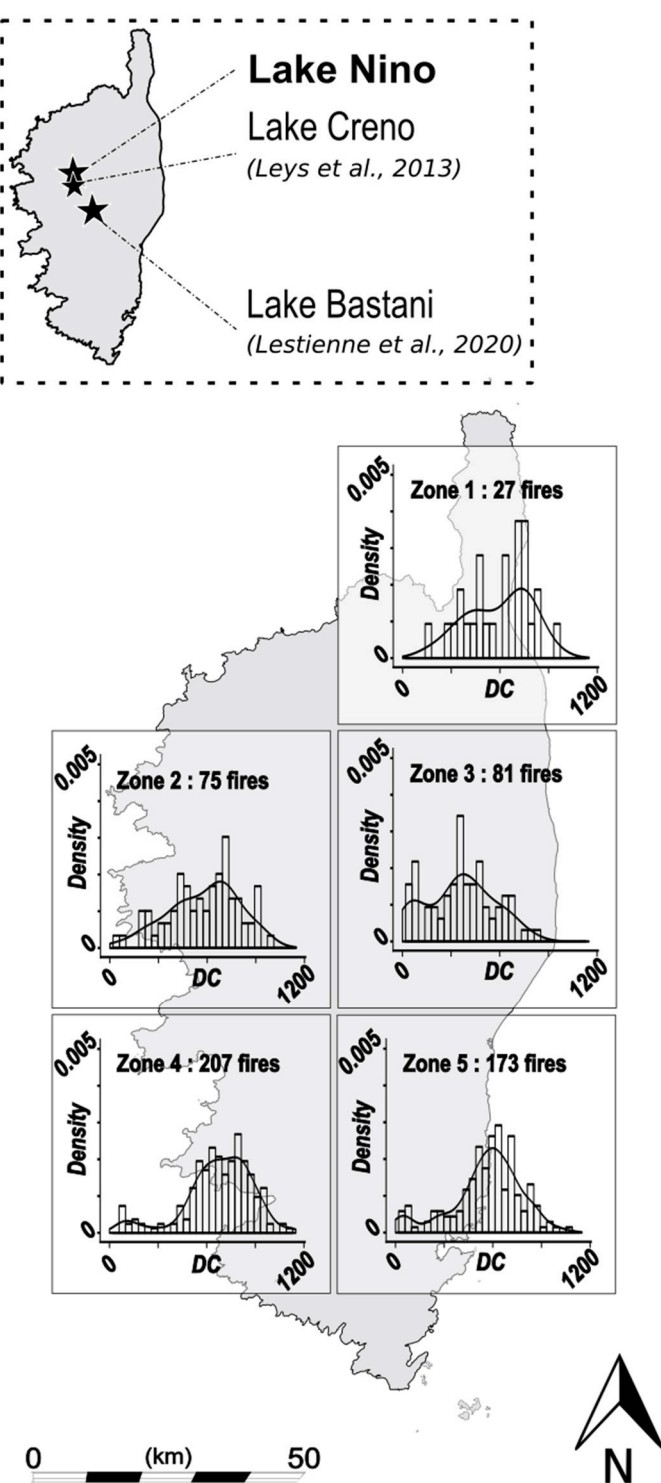

**Figure A2.** The high spatial variability in terms of fire occurrences reported since 1979 in Corsica (the southern region recording more fires than the northern part [10]) and the related distribution of the DC values for fire-days from the five ERA-Interim pixels covering Corsica.

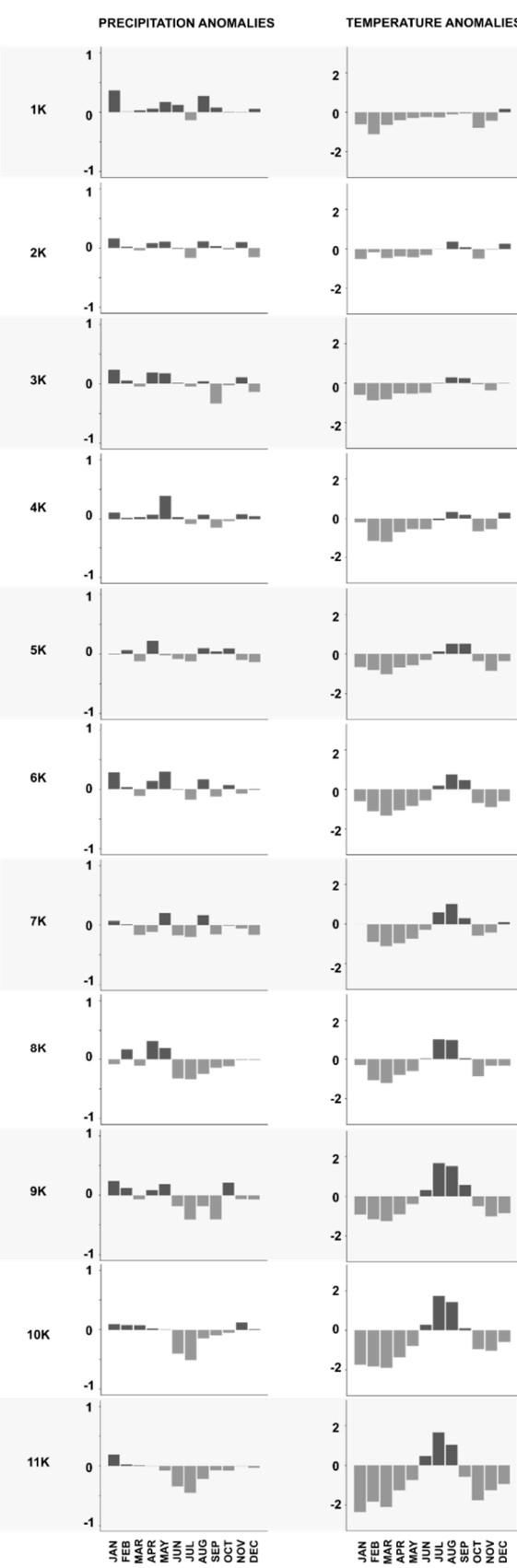

**Figure A3.** Precipitation and temperature anomalies during the Holocene calculated from the HadCM3B-M1 dataset [52].

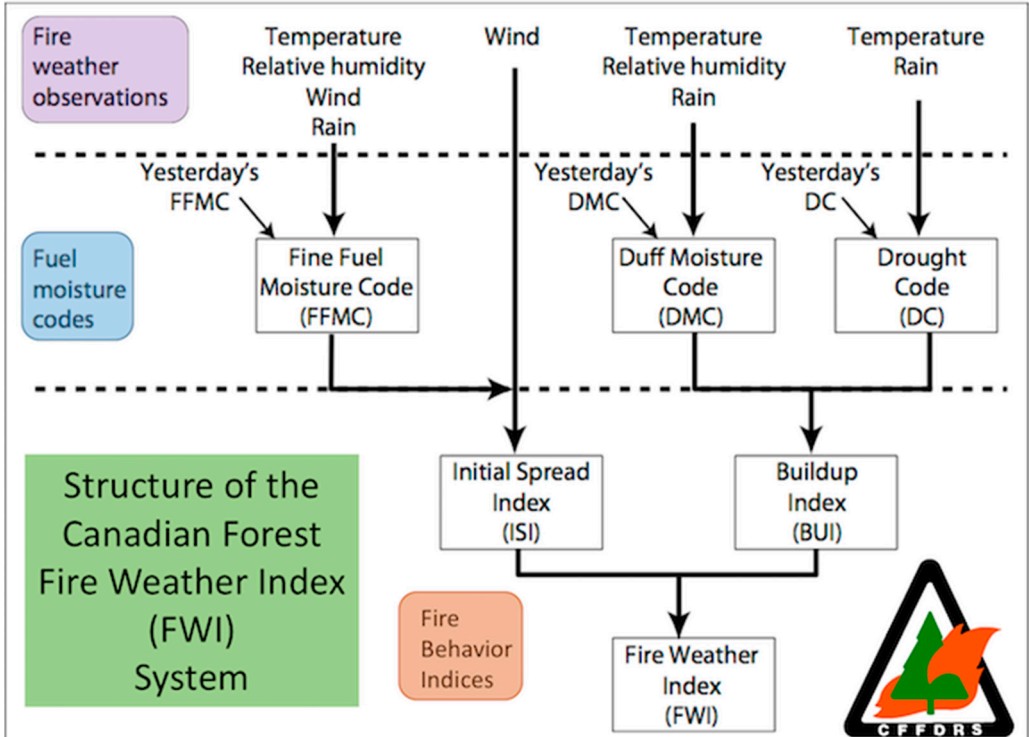

**Figure A4.** Fire weather index calculation scheme [23] and DC computation code.

Daily computation of:
Potential Evapotranspiration (PE):

$$PE = (0.36 \times (T + 2.8) + Lf$$

where T represents the temperature and Lf represents the seasonal adjustment of day length [23].
Daily Humidity index (DH):
if there is no rain (or rain ≤ 2.8 mm), then:

$$DR = DC_0$$

where $DC_0$ is the DC value for the previous day;
if there is rain (rain > 2.8 mm), then:

$$DR = DC_0 - 400 \times \log(1 + 3.937 \times RW/SMI)$$

where RW is the rain and SMI is the humidity index for the previous day.
Finally, Daily DC (DC) = DH + PE.

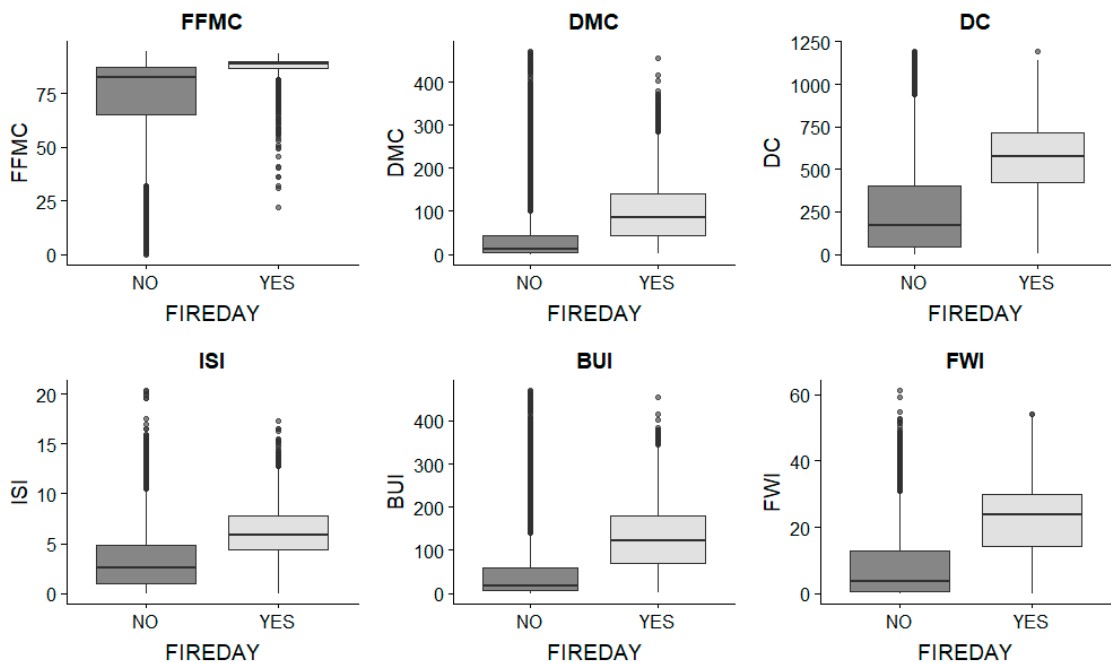

**Figure A5.** Comparison of Fire Weather Indexes (FWIs) as a function of fire-day status (YES with fire reported, NO without fire reported). *p*-values are < 0.001*** for all indexes (*t*-test with resampling to have groups with the same size).

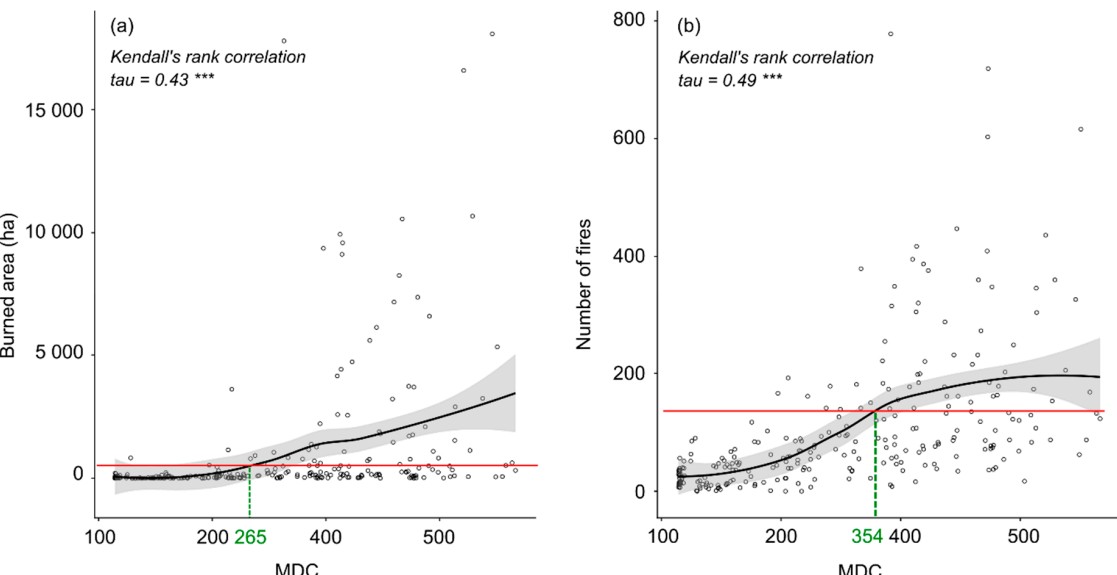

**Figure A6.** Relationship between MDC and total burned area (**a**) or the number of fires (**b**) between April and October over the 1979–2016 period. Each trend was calculated using the LOESS method. In both panels, the red line represents the most fire-prone months (fourth quartile), while the green line points to the corresponding MDC value. These "local threshold values" corresponding to the most fire-prone months were 265 and 354 (for burned area and number of fires, respectively), averaging 310 units, which is very close to the 300 unit threshold value estimated from DC.

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
