# Peer review of "Combining the Monthly Drought Code and Paleoecological Data to Assess Holocene Climate Impact on Mediterranean Fire Regime"

_fire, doi:10.3390/fire3020008_

Round 1
Reviewer 1 Report
I am very sorry to again recommend significant revision of this manuscript. Much of my review is a repetition of my previous comments. As far as I can determine, the critique made in the previous review with respect to the application of the CharAnalysis program was ignored. In fact the discussion if the CharAnalysis methodology is reduced in this version. As such I will modify this review only slightly.
This study uses the Monthly Drought Code (MDC) to reconstruct fire season length for Corsica over the last 11 ka years BP. The impetuous for this is the availability of monthly mean and maximum climate data from climate simulation models extending to periods when daily weather data are not available. Validation of the efficacy of this approach is demonstrated by the strong correlation between the DC and MDC values, as well as the length of the fire season, calculated for the calibration period of 1979-2016. Independent proxies for climate and land use are used to inform and validate the interpretation of the MDC reconstructions. Pollen is used to reconstruct vegetation cover and the presence of agriculture, Sporormeilla sp. is used an indicator for the presence of domesticated herds, and charcoal is used as a proxy for fire events. The authors interpret their results as showing a clear pattern of longer fire season length between 11 ka and 7 ka BP, followed by a decrease more or less stabilizing until the last millennium BP. The primary driver of fire regime shifts from climate to human control at about 5 Ka BP.
The aspects of this study involving the use and validation of MDC for reconstruction of fire season are well conceived.
The use of the independent proxy evidence for vegetation and grazing are also well presented and logically argued.
The presentation, analysis, and interpretation of the charcoal data is, however, flawed. Following is a list of critiques:
The discussion of the CharAnalysis algorithm and method in this version is reduced relative to the previous submission. At minimum what is needed:
- The parameters used in CharAnalysis need to be listed, including the smoothing algorithm chosen, the confidence interval designated, the method for calculating background (residual or index).
- The signal to noise index needs to be show for the entire record for each site in order to demonstrate a SNI of greater than 3 at minimum.
- The widow width chosen for peak identification is not discussed and the current version. In lines 127-131 there is only brief reference to the the CharAnalysis model. Lines 158 – 160 in the previous submission stated: “cores have been resampled by … (20 years for Nino and Creno lakes, 100 years for Bastani lake). And a moving time window of 500 years has been used…”. This would mean that for Nino and Creno lakes the number of interpolated values included in each window is 25, and for Bastani only 5. The minimum number of values needed for statistically significant results is 30 per window (Higuera et al. 2010). This is a major error in the use of CharAnalysis. If there has not been a change in the window widths used, the analysis must be redone before publication of the results. And this must be discussed in the methods and results.
The continued absence of the presentation of these factors in the employment of CharAnalysis leads to the impresson that the authors are treating the program as a "blackbox" and do not understand the method they are utilizing. I suggest the authors re-read the paper by Higuera et al. 2010:
Higuera, P., Gavin, D., Bartlein, P., Hallett, D.J., 2010a. Peak detection in sediment–charcoal records: impacts of alternative data analysis methods on fire-history interpretations, International Journal of Wildland Fire, pp. 996-1014.
Let me be clear: I am very intrigued by the study in terms of the innovative and thoughtful way in which the authors have applied the methodologies they employ. The organization and logical flow of the manuscript is very well done. As stated previously, I would be in support of publication on the condition of revisions addressing the issues raised with respect to the application of CharAnalysis. Specifically, as stated in my last review: Reanalysis and more comprehensive presentation of the charcoal data, CharAnalysis methodology, analysis parameters, and results. If the records being analysed to not support the application of CharAnalysis due to a lack of sampling resolution, the authors might consider looking at changing background values in charcoal influx as an alternative to peak detection.
Author Response
We sincerely apologize for the confusion in our edits in this second version of the manuscript, in particular for those edits which were related to the supportive suggestions and questions of reviewer #1. Indeed, we decided to remove all charcoal-data analyses done with the CharAnalysis software in this second version of the manuscript to be consistent with the main objectives of the paper: introduction of the MDC and FSL for the Mediterranean and Corsica region in particular, MDC and FSL estimation of the millennial values for the Holocene, and comparison with the three Holocene charcoal records available in Corsica including an original one. The charcoal-peaks detection has been already deleted since the last submission from the figures. But we forgot to remove a few sentences related to the CharAnalysis methodology and we have not been clear enough about the resampling and the rescaling. All three data-series have been resampled at the same time-resolution to make them comparable among each other without any more signal treatment related to charcoal peak detection as CharAnalysis does. In a other hand, a trend curve was generated from the rescaled CHAR values of the three lakes (Bastani, Nino, and Creno) using the loess (locally estimated scatterplot smoothing) method. To explain our strategy, we add a few words (l175-180) to explain the methods, the bin value of 20 years and how we drew the trend over the documented period in the text (see figure 4). We hope these corrections, clarifications and edits answered adequally to reviewer #1 comments and requests.

Reviewer 2 Report
General comments:
The manuscript by Lestienne et al. titled ‘Combining the Monthly Drought Code and Paleoecological data to assess the Holocene climate impact on Mediterranean fire regimes’ presents an interesting study including the promethee fire database, climate reanalysis and model results with charcoal records of fire activity and pollen indicators of human activity from lake sediments.
The manuscript clearly developed during the revision process. However, before publication in Fire I would still like to see a few more details on the methodology and, more important, a discussion on the changing relevance of the Monthly Drought Code (MDC) during the calibration period back to 1979, compared to the Holocene.
Specific comments:
DC computation:
Add a formula for the computation of DC values in the main text.
MDC computation:
From the text: Then, the MDC values have been computed for the entire Holocene (0 to 11 Ka BP) based on the HadCM3B-M1 climate datasets in order to compare the MDC trend with the paleofire trend from charcoal records.
Maybe replace: ‘climate’ with ‘temperature and precipitation’.
MDC relevance:
In contrast to the general ability of the MDC to capture fire occurrences back to 1979, a period with strong human disturbances and climate variability, the index during the Holocene ‘only’ works as a drought index. This is now considered in the text and title. However, in addition, I would like to see some discussion on the important change in proxy relevance and, in particular, its implications for the calibration results.
Author Response
General comments:
The manuscript by Lestienne et al. titled ‘Combining the Monthly Drought Code and Paleoecological data to assess the Holocene climate impact on Mediterranean fire regimes’ presents an interesting study including the promethee fire database, climate reanalysis and model results with charcoal records of fire activity and pollen indicators of human activity from lake sediments.
The manuscript clearly developed during the revision process. However, before publication in Fire I would still like to see a few more details on the methodology and, more important, a discussion on the changing relevance of the Monthly Drought Code (MDC) during the calibration period back to 1979, compared to the Holocene.
Cf below
Specific comments:
DC computation:
Add a formula for the computation of DC values in the main text.
Done , we have added an explanation of the DC equation in lines 241 – 252, and we have added the fire Weather Index calculation scheme in appendix F.
MDC computation:
From the text: Then, the MDC values have been computed for the entire Holocene (0 to 11 Ka BP) based on the HadCM3B-M1 climate datasets in order to compare the MDC trend with the paleofire trend from charcoal records.
Maybe replace: ‘climate’ with ‘temperature and precipitation’.
Done. The revised sentence is now on line 281.
MDC relevance:
In contrast to the general ability of the MDC to capture fire occurrences back to 1979, a period with strong human disturbances and climate variability, the index during the Holocene ‘only’ works as a drought index. This is now considered in the text and title. However, in addition, I would like to see some discussion on the important change in proxy relevance and, in particular, its implications for the calibration results.
Both DC and MDC are only weather indexes without any consideration of human influence. They both capture the moisture in deep humus layer, which can been translated into large dead woody debris (branches) on the ground. Their time lag to dessicate or wet in so long (52 days from full saturation under laboratory controled conditions) that they are assimilated to a proxy of drought through the season. Therefore, they present the same meaning today and back during the Holocene, and the same proxy value or strength through time.
We explicitely added the following sentence for this on new line 266-268 : “ The fact that there is no human activity related to the DC computation is a valuable characteristics that allows us to use it to assess past periods such as earlier in the Holocene or before”.
We also added a part in the limits (l519-523): “If we have seen that the DC was able to discriminate fire days, and that the MDC can discriminate fire-prone months, it is indeed important to understand that today's fires are mostly linked to human activity. In this logic, it is likely that the threshold value chosen on the basis of current data overestimates the fire danger over the oldest periods.”

Reviewer 3 Report
Line 224: At that time, FS both started later and ended later. I wonder if you intended to write "ended earlier".
Author Response
Comments and Suggestions for Authors
Line 224: At that time, FS both started later and ended later. I wonder if you intended to write "ended earlier".
Yes sorry for this. The revised sentence is now on line 363-364.

Round 2
Reviewer 1 Report
I am in full support of publication of this work.
The integration of different complementary modeling methods is intriguing and very instructive. The authors have wrangled multiple data sets into a form that is cohesive and coherent, resulting in an elegant study validating the reliability of the predictive models based on the longer term view afforded by the paleo record.
This type of work is just the sort of practical application and integration of paleo and modern perspectives that is most pertinent to our urgent need to understand and anticipate the ecological dynamics of our current, rapidly changing, climatic and environmental context.

Author Response
We thank the reviewer for these corrections and for his time.
The terms have been modified according to the reviewer's advice in the new lines 179-182:
" Based on the sample-age average, the cores were resampled by 20 years (approx. the mean step of Nino and Creno cores) to make them comparable. A smooth curve was generated from the resampled and rescaled (Z-score) CHAR values of the three lakes (Bastani, Nino, and Creno) using the loess (locally estimated scatterplot smoothing) method."
We have also replaced the term "global fire signal" with "Smoothed regional signal" which is indeed more appropriate.
This manuscript is a resubmission of an earlier submission. The following is a list of the peer review reports and author responses from that submission.
Round 1
Reviewer 1 Report
Thorough experimental and calculative work which points out the value of Monthly Drought Code (MDC) as a valid fire danger indicator for the past (Holocene) and present eras of the Mediterranean region. Regional paleo-fire reconstructions were done by analyzing the charcoal content in sediments of one Corsican lake, while data from 2 other lakes were derived from the literature. The carbon peaks (indicating fire incidents) were correlated with calculated values of MDC through the Holocene, thus suggesting (aided also by biological markers) whether the fire origin was climatic or anthropogenic. The article is well articulated. Thus, I only have minor comments.
Throughout:
Figure and Table should not be abbreviated to Fig. and Tab. Introduce a space, where missing, between a number and its units. Add section numbers, e.g. 2.1. Study Area and Sampled Lakes and capital letters in the start of the words. Check that all abbreviations are explained when first used Some minor adjustments to the English writing are needed Figure 4: Normal chronology? Start at -12KY to the left and finish with 0 to the right? This will make it easier for non-paleo-ecologists to read the figure. (Leys et al. (2013) follow the suggested chronology.)
Abstract: The content mentioned in lines 67 – 70 should also be included in the abstract.
Line 19: Daily climate-data requirement, consider daily weather-data requirements (applies also to line 58).
Line 32: Consider whether "Promethee" shall have accents or not (vs. line 107)
Line 51: fuel moisture content
Line 69: In my opinion the phrase "based on sedimentary charcoal contents from Three Corsican lacustrine records" should also be mentioned in the abstract. The fact that experiments were performed in Lake Nino in the present work; while it is referred to the literature for the results from the 2 other lakes should also be mentioned in the abstract and in the introduction.
Line 80: 2.1 Study area and sampled lakes
Line 81: Consider Space between 80 and km. (Same for line 82)
Line 85: Check whether the abbreviation BP is explained
Lines 93 to 105: Decide if you will write Lake (recommended) or lake for all Three Lakes.
Figure 1: Lake Nino is in bold face, because the present work produced the data (I guess). Lake Creno is not in bold face, and a reference is given. Lake Bastani is also in bold face, and a reference is not given in the Figure. This is perhaps because the article, (with the same first author as the present one) though accepted, is not yet published. This introduces some inconsistency in the Figure. The comment applies also for Figure 4.
Line 106: 2.2 Current fire and climate datasets
Line 118: 2.3 Charcoal, pollen and fungal remain analysis
Line 119 – 121: Lake Creno is not mentioned here, vs. lines 93 – 105?
Line 132 and 133: Arboreal Pollen Grains, or grains?
Line 138: 2.4 Climatic model
Lines 144 – 145, perhaps refer to Appendix D?
Line 151: Consider space between number and unit
Line 153: 2.5 Fire Weather Index System
Line 155: Table 2
Line 155 and Table 2: Check whether Monthly Drought Code is among the sub-indexes of the Canadian Fire Weather Index System
Line 169: Do you mean “respective period fire signal”?
Line 173: 2.6 Fire Season (do not introduce the abbreviation in the section title)
Line 183: 3.1 DC and MDC …
Lines 185 and 188: Figure 2
Line 194: ***?
Line 202: Figure 3a, Figure 3b
Figure 3: check that you follow the template. Perhaps remove capital letters A and B from the top of the figures, and replace them with (a) and (b) under the figures. Check that it then is consistent with the figure text.
Line 213: 3.2 Change in the …
Line 215: Figure 4
Line 219: Figure 4, Table 3
Line 219: Fire Season Length (FSL)
Line 224: from July to September
Line 235: 3.3 Holocene …
Lines 236, 240, 258: Figure 4
Line 264: 4.1 The …
Line 282: 4.2 Holocene
Line 285: Figure 4
Line 296: This strengthens the idea
Line 323: … to suggest that the main driver …
Line 327: … to increased fire frequency
Line 338: 4.3 Current …
Line 340: Our results showed / Our results confirmed?
Line 370: Consider to move the word “reached” between millennia and values.
Line 377: remove “of”.
Reviewer 2 Report
General comments
This study used historic fire records from Corsica and ERA-interim reanalysis climate data to identify thresholds in the Drought Code (DC) that separate fire days from non-fire days, and then applies the same DC threshold to simulated Monthly Drought Code (MDC) for the Holocene to make predictions about fire activity and the length of the fire season during the Holocene. Independent records of fire activity, vegetation composition, and human occupancy are reconstructed using charcoal and pollen records from three lakes in Corsica. I have major concerns about the MDC, its interpretation and its application to paleo records. In this study 1979-2016 fire records from the island of Corsica (including rest of Southern France?) have been used to identify the (daily) Drought Code threshold value (DC) that best separated fire days and non-fire days in the record. This daily DC threshold (DCfire) is then assumed to apply equally well to the Monthly Drought Code (MDC). This step is justified with Figure 3, which shows that the monthly mean of daily DC (DCmean) is highly correlated to the MDC calculated directly from monthly climate data following ref 24 (Girardin & Wotton, 2009). This justification is incomplete and therefore unconvincing. The fact that DCmean and MDC are highly correlated is to be expected, but that does not mean that MDC is a reliable predictor of the probability that daily DC values in that month exceeded DCfire or a reliable indicator of the frequency of fire days in that month. To use this approach, one would need to find the MDC value that best separates months with fire from those of no fire, or a relationship between MDC and P(fire). The second (and related) concern I have with this study is the lack of validation showing that DCfire, as derived from recent fire and climate records, is robust for the very substantial variation in climate and vegetation seen on the island over the Holocene (Fig 4). Assuming that the DC is an index of surface fuel moisture content (which has been shown to be highly uncertain, e.g. see Schunk et al. 2017, Agricultural and Forest Meteorology 234) then we interpret the empirical DCfire value as indicative of the DC value at which the current vegetation on the island shifts from a humid, non-flammable, fuel state to a dry, flammable, fuel state. This empirical DCFire lacks the biophysical basis to allow it to reliably predict P(fire) across different combinations of vegetation type and climate. As indicated by the variation in the AP/NAP ratio of the pollen record (Fig. 4), the vegetation cover during the Holocene has varied between a dominance of herbaceous species and dominance of woody species, and between broadleaf and coniferous tree species. This variation in vegetation composition and fuel types can be expected to be associated with different DCfire thresholds. Similarly, changes in general circulation patterns and climate during the Holocene could also cause DCfire to be quite different from the one based on 1979-2016 fire and climate data, even if the vegetation cover would have been constant over the entire study period.
Specific comments
Title: I don’t think the claim in the title is supported by the actual findings of this study. See general comments above. 42-48: The question whether climate or humans drive fire regimes is ill-posed. Fire regimes emerge from climate-vegetation-fire-human interactions operating at a range of time scales. See e.g. Bradstock (2010). Global Ecology and Biogeography 19(2). 109: it is not clear what fire data have been used. Only fires recorded on Corsica or all fires recorded in Southern France? Table 1: there must be an error in the burned area data, as the median fire size in Corsica is unlikely to be 100,000 ha. Figure 4: consider using colour in the pollen diagram. 265-268: This claim about general use of the DC is not supported by the findings of this study. See Schunk et al. 2017, Agricultural and Forest Meteorology 234, for a validation of FWI moisture indices against actual fuel moisture measurements. 299-303: Given such pronounced changes in vegetation composition and associated fuel properties, you can’t assume that the DCFire you found empirically for current vegetation and climate conditions will be a reliable indicator of P(fire) throughout the Holocene. 351-353: I don’t agree with this claim. You have not tested the efficiency of the MDC to discriminate fire days from non-fire days. Rather you have found a daily DC that discriminates fire days from non-fire days under current climate and vegetation, which is not the same. See general comments above. 361-362: you have not shown that the MDC is an efficient tool. See previous comments.
Reviewer 3 Report
I am very interested in this study, however I felt that I was not provided with a thorough enough discussion of the methodology to offer a critique of the work. The authors description of the CharAnalysis program does not reveal their understanding of the algorithm. Nor do the authors reasure the reader of the appropriateness of their charcoal data for interpretation using CharAnalysis. I would like to know what the original charcoal concentrations were (raw data) and how the cores were sampled (increments of sub samples down core, volume sampled). Of significant concern is the authors failure to reference important aspects of interpreting CharAnalysis results such as the SNI, confidence intervals, and widow widths. I also need to know how the automated counting was done. Both of these are referenced in the paper, but that paper (Lestienne et al. submitted) is not available. And, the reader should not have to do a literature search to evaluate the methods for the study presented in the current paper. This also applies to the methods used in defining the Fire Weather Index System and the Drought Code and the Monthly Drought Code and the Fire Season. All of these methods are referenced, but they need to be explained for the reader in this manuscript. The burden of the effort to understand the methodology sufficiently to 1) critique the methods and 2) to replicate the study has been left too much to the reader.
I would be pleased to review a revised version of this paper which includes a more thorough treatment of the methods and of the original charcoal data (sampling and counting methodology and raw count values)
Reviewer 4 Report
General comments:
The manuscript by Lestienne et al. titled ‘The Monthly Drought Code: a tool adapted to the Mediterranean region to discriminate the climatic and human origins of fires’ presents a study including the promethee fire database and climate reanalysis data as well as fire reconstructions from lacustrine sediments and pollen indicators of human activity. Efforts undertaken aim at providing a novel index for the reconstruction of Holocene wildfire activity for the Mediterranean region, independent from existing daily measurements. However, while large parts of the manuscript suggest that the Monthly Drought Code (MDC) is an index of wildfire activity, it is ‘only’ a climate index that cannot explain wildfire occurrences on its own. Therefore, parts of the manuscript should be in reformulated and the multi-proxy approach to explain wildfire activity better clarified before publication in ‘Fire’ (see my main comment).
Main comment MDC:
In contrast to the suggestions in the title, abstract and conclusions, MDC is not a standalone index of wildfire activity. MDC is a climate index and further proxy records of (in this case) human activity are needed to interpret changes in wildfire occurrences as shown in the charcoal records. I agree that this information is to a degree provided in the text, but the general sound of the manuscript appears different. Therefore, keeping this in mind, the manuscript should be in parts reformulated.
Specific comments:
(1) Abbreviations:
I would reduce the number of abbreviations in the text, when applicable. The large number of abbreviations are difficult to remember and complicate the reading.
(2) Construction of the MDC index:
Provide more details on the calculation of the MDC index and how the connected unit can be interpreted. How dry is, for example, a MDC of 400?
(3) KY
KY alone does not specify a time. 9 KY means 9000 years and not a point in time. Okay, the abbreviation KY is defined in Line 100. Maybe ‘ka BP’ is easier to understand?
(4) Lines 27-28 (unprecedented):
MDC values slightly increase today (within errors they do not increase at all during the last 7000 years and they are lower compared to 8-12 ka BP). Therefore, the expression ‘unprecedented values since 2 KY’ is wrong.
(5) Climate model forcing (line 142):
Is the model only driven by changes in orbital forcing? What about ocean circulation, volcanic forcing, solar activity?
Minor comments:
Line 30 and throughout the text: Delete ‘dangerous’ or specify what that means.
Line 43: Define the term ‘fire weather’.
Line 63: Correct ‘future’.
Line 72: Correct ‘terms’.
Line 86: Replace ‘most’ by ‘remaining’.
Lines 93-105: Provide coordinates also for Lake Bastani, like for the two other sites.
Line 104: Replace ‘to’ with ‘with’.
Line 110: I guess ‘0.5 ha’ should be ‘5 ha’?
Line 174: Correct ‘developed’.
Line 194: What do the three stars stand for?
Line 282: Replace ‘anthropic’ with ‘anthropogenic’.
Line 285: Format ‘Fig. 4’.
Line 338: Delete ‘dangerously’?
Line 341: Replace ‘beyond’ with ‘below’.
Figure 1: Explain why the name of some lakes are in bold while others are not. Add the reference for Lake Bastani to the figure, like for Lake Creno.
Round 2
Reviewer 2 Report
General comments
In the review of the original manuscript, I raised two major issues of concern: i) the use of daily (DC) and monthly drought code (MDC) to identify a threshold DC for fire occurrence in Corsica (DCfire), and ii) the assumption that DCfire is robust for the very substantial changes in climatic and other environmental conditions over the Holocene. Neither issue has been adequately addressed in the revised manuscript, so I will reiterate my points here:
My understanding is that firstly (FIGURE 2) daily DC values (1979-2016) for five large grid cells covering Corsica were analysed to identify a threshold DC that best separates fire days from non-fire days (DCfire, a value of 405 was found). Secondly, the case was made (FIGURE 3) that the monthly mean of daily DC values (DCmean) is highly correlated with the monthly drought code (MDC). And thirdly, the DCfire threshold was applied to modelled MDC for the Holocene to help interpret the fire proxy time series of charcoal fragments in lake sediments. The monthly mean of daily DC is unlikely to provide a good estimate of the days with DC above or below DCfire. Instead, as outlined in the previous review, the analysis should have focused on quantifying the relationship between monthly DCmean (monthly mean of daily DC) and the monthly frequency or probability of a fire day. This can easily be done with the data you already have. That would resolve my first point of concern.
My second point of concern is not easily fixed with the existing data, but should at least be thoroughly addressed in the discussion. Like any statistical model relating weather of climate to fire activity in a particular region over a particular observation period, the threshold DC value identified in this study cannot be readily applied to other environments or times with fundamentally different climate/weather/vegetation/fuel conditions because you don’t know that the model is robust for such changes. In this study, DCfire is an empirical threshold identified in the fire-weather data for contemporary Corsica with a particular climate and (spatial pattern of) vegetation cover and land use. From basic fire science one would expect DCfire to be fairly sensitive to variation in climate, vegetation cover or land use. Therefore, it is very unlikely that the same DCfire would apply throughout the Holocene with the dramatic changes in climate, vegetation and land use you have identified. The current manuscript lacks a convincing justification for this assumption.
Reviewer 3 Report
This study uses the Monthly Drought Code (MDC) to reconstruct fire season length for Corsica over the last 11 ka years BP. The impetuous for this is the availability of monthly mean and maximum climate data from climate simulation models extending to periods when daily weather data are not available. Validation of the efficacy of this approach is demonstrated by the strong correlation between the DC and MDC values, as well as the length of the fire season, calculated for the calibration period of 1979-2016. Independent proxies for climate and land use are used to inform and validate the interpretation of the MDC reconstructions. Pollen is used to reconstruct vegetation cover and the presence of agriculture, Sporormeilla sp. is used an indicator for the presence of domesticated herds, and charcoal is used as a proxy for fire events. The authors interpret their results as showing a clear pattern of longer fire season length between 11 ka and 7 ka BP, followed by a decrease more or less stabilizing until the last millennium BP. The primary driver of fire regime shifts from climate to human control at about 5 Ka BP.
The aspects of this study involving the use and validation of MDC for reconstruction of fire season are well conceived.
The use of the independent proxy evidence for vegetation and grazing are also well presented and logically argued.
The presentation, analysis, and interpretation of the charcoal data is, however, flawed. Following is a list of critiques:
The original count data, by level, needs to be shown for each core, not the total number of charcoal particles per core counted (line 151) The parameters used in CharAnalysis need to be listed, including the smoothing algorithm chosen, the confidence interval designated, the method for calculating background (residual or index). The signal to noise index needs to be show for the entire record for each site in order to demonstrate a SNI of greater than 3 at minimum. The widow width chosen for peak identification is too small to support significance in the results: lines 158 – 160 “cores have been re-sampled by … (20 years for Nino and Creno lakes, 100 years for Bastani lake). And a moving time window of 500 years has been used…”. This would mean that for Nino and Creno lakes the number of interpolated values included in each window is 25, and for Bastani only 5. The minimum number of values needed for statistically significant results is 30 per window (Higuera et al. 2010). This is a major error in the use of CharAnalysis. If this is not a misrepresentation of the window widths used, the analysis must be redone before publication of the results.
I am very intrigued by the study in terms of the innovative and thoughtful way in which the authors have applied the methodologies they employ. The organization and logical flow of the manuscript is very well done. I would be in support of publication on the condition of revisions addressing two issues:
Reanalysis and more comprehensive presentation of the charcoal data, analysis parameters, and results as indicated above. Significant improvement of the english language is needed. Through out the paper the language syntax and vocabulary frequently make it very difficult to understand what the authors are saying. Spelling also deserves a bit more attention. I strongly recommend editing by an english language editor before publication.
Higuera, P., Gavin, D., Bartlein, P., Hallett, D.J., 2010a. Peak detection in sediment–charcoal records: impacts of alternative data analysis methods on fire-history interpretations, International Journal of Wildland Fire, pp. 996-1014.
Reviewer 4 Report
I am happy with the authors' revisions. Please check the text for some English mistakes. Thank you.